



# Contributions of foreign, domestic and natural emissions to US ozone estimated using the path-integral method in CAMx nested within GEOS-Chem

Alan M. Dunker[1], Bonyoung Koo[2], Greg Yarwood [2]

[1]A. M. Dunker, LLC, 727 Robinhood Circle, Bloomfield Hills, MI 48304, USA
[2]Ramboll Environ, 773 San Marin Dr., Suite 2115, Novato, CA 94945, USA

*Correspondence to*: Alan M. Dunker (alan.m.dunker@gmail.com)

**Abstract.** The Goddard Earth Observing System global chemical transport (GEOS-Chem) model was used at 2° x 2.5° resolution to simulate ozone formation for a base case representing year 2010 and a natural background case without worldwide anthropogenic emissions. These simulations provided boundary concentrations for base and natural background simulations with the Comprehensive Air Quality Model with Extensions (CAMx) on a North American domain at 12 km x 12 km resolution over March–September 2010. The predicted maximum daily average 8-hour (MDA8) background ozone for the US is largest in the mountainous areas of Colorado, New Mexico, Arizona, and California. The background MDA8 ozone in some of these locations exceeds 60 ppb, when averaged over the 10 days with the largest base-case ozone. The ozone difference between the base and background cases represents the increment to ozone from all anthropogenic sources. Using the Path-Integral Method, the anthropogenic ozone increment was allocated to US anthropogenic emissions, Canadian/Mexican anthropogenic emissions, and the anthropogenic components of the lateral and top boundary concentrations (BCs). For the larger MDA8 ozone concentrations in the base case, the relative importance of the sources is generally US emissions > anthropogenic lateral BCs > Canadian/Mexican emissions >> anthropogenic top BCs. The contributions of the lateral BCs are largest for the higher elevation US sites in the Intermountain West and sites closest to the boundaries. If the focus instead is on the larger ozone concentrations in the background case, the contribution from US emissions is reduced leading to a reduction in the anthropogenic ozone increment. The contribution of the Canadian/Mexican emissions remains about the same, and the contribution from the lateral BCs increases at lower elevation urban sites. The net effect is that the relative importance of the anthropogenic lateral BCs is significantly increased for the days with the largest background concentrations. In addition to the source apportionment, we also used surface and ozonesonde measurements to evaluate GEOS-Chem and CAMx performance.

## 1 Introduction

In 2008, the US Environmental Protection Agency (EPA) reduced the National Ambient Air Quality Standard (NAAQS) for 8-hour ozone ($O_3$) to 75 ppb, and in October 2015, further reduced the NAAQS to 70 ppb. An important consideration is





how difficult it will be to meet this standard by reducing US emissions alone. One issue is the magnitude of the $O_3$ background in the absence of anthropogenic emissions. Another issue is the contribution of anthropogenic emissions outside the US to US $O_3$. This can occur by the transport of the foreign anthropogenic emissions but more importantly by the transport of the foreign $O_3$ and other secondary pollutants.

Background $O_3$ must be estimated from model simulations, though a related quantity, termed baseline $O_3$, can be estimated from data at relatively remote monitoring sites (Parrish et al., 2012). McDonald-Buller et al. (2011) review background and baseline $O_3$ and how the latter can be used to test model results. There are three common definitions of background $O_3$: US background, North American Background and natural background, which correspond to elimination of US, North American or worldwide anthropogenic emissions, respectively (Zhang et al., 2011; Emery et al., 2012; Fiore et al., 2014; Zhang et al.,

2014; Dolwick et al., 2015; Nopmongcol et al., 2016). The anthropogenic increment of $O_3$ is the difference between a base-case simulation with all emissions present and the chosen type of background simulation.

One approach to apportioning the anthropogenic increment is to remove sources one at a time and determine the change in $O_3$ from the base case (brute-force or zero-out method) (Zhang et al., 2014; Dolwick et al., 2015). A limitation is that the sum of all the anthropogenic source contributions generally does not equal the anthropogenic increment due to the nonlinear

chemistry. Another approach is to add reactive tracers (tagged species) to the base case for the emissions from anthropogenic sources and/or the secondary pollutants formed from the emissions and then use the tracers to estimate the contribution of the sources to the total $O_3$ concentration (Zhang et al., 2008; Lefohn et al., 2014; Baker et al., 2015; Dolwick et al., 2015; Nopmongcol et al., 2017). The chemistry causes interactions between the sources through direct and indirect effects. For example, if $O_3$ (generated from anthropogenic emissions) is transported into the domain through the boundaries,

direct effects are destruction of the $O_3$ by reaction with anthropogenic NO emissions in the domain, $O_3 + NO \rightarrow NO_2 + O_2$, or reaction with $HO_2$ formed from volatile organic compound (VOC) emissions, $O_3 + HO_2 \rightarrow OH + 2 O_2$. Indirect effects are the photolysis of the $NO_2$ to recreate $O_3$ and the reaction of the OH with anthropogenic VOC emissions to form $O_3$. Reactive tracers can follow some indirect effects (Baker et al., 2015; Nopmongcol et al., 2017) but ultimately not all the nonlinear chemical interactions among emissions from different sources can be included. In addition, there is no requirement that the

sum of tracer contributions ascribed to the anthropogenic sources equals the anthropogenic increment (as defined above). Also, the focus in the tracer approach is solely on the chemistry in the base case.

The Path-Integral Method (PIM) for source apportionment has the unique capability to allocate the difference in $O_3$ between two simulations (e.g., the anthropogenic increment) to portions of the emissions and boundary concentration (BC) changes between the two simulations (Dunker, 2015; Dunker et al., 2015). The PIM determines the source contributions by

30 integrating first-order sensitivity coefficients over the range of emissions from the background case to the base case, and thus the source contributions are not determined just from the chemistry in the base case. Calculating the sensitivity coefficients involves the same Jacobian matrix used in solving the chemical reaction equations. Consequently, the source contributions implicitly include all the direct and indirect effects represented by the chemical mechanism. The sum of the source contributions is constrained to equal the anthropogenic increment, within the accuracy of the numerical integration. The





disadvantage of the PIM is that it requires more computational effort than the brute-force or tracer methods. Dunker et al. (2015) determined the anthropogenic increments in $O_3$ and other species defined by the difference between a base case for 2030 and the US background and allocated the increments to US source categories using the PIM.

For this study, we used the Comprehensive Air Quality Model with Extensions (CAMx; Ramboll-Environ, 2016) in a one-
5 way nest within the Goddard Earth Observing System global chemical transport (GEOS-Chem) model (Bey et al., 2001) to estimate the anthropogenic increment in $O_3$ equal to the difference between a 2010 base case and the natural background. Using the PIM, this increment was then allocated to the US and Canadian/Mexican anthropogenic emissions within the CAMx North American modeling domain and to worldwide anthropogenic emissions outside the domain that impact the CAMx simulation via the BCs. We also conducted an evaluation of the CAMx and GEOS-Chem model performance for the
10 base case using various surface and ozonesonde measurements.

## 2 Methods

### 2.1. GEOS-Chem simulations

Two global model simulations were conducted to provide BCs for North American regional model simulations using the latest version of GEOS-Chem available for this work, version 10-01 (http://www.geos-chem.org). The base-case global
simulation (G-Base) used global anthropogenic emissions from the Emissions Database for Global Atmospheric Research (EDGAR v4.2) with anthropogenic volatile organic compound (VOC) emissions from the Reanalysis of the Tropospheric chemical composition (RETRO) emission inventory. Additional global databases provided anthropogenic emissions from aircraft, ship, fertilizer, and biofuel sources. Regional anthropogenic emission inventories superseded the global emissions for specific regions of the world, e.g., the US, Canada, Mexico, Europe, and Southeast Asia. The natural emissions included
biogenic emissions, biomass burning, nitrogen oxides ($NO_x$) from lightning and soil, volcanic emissions, and wind-blown dust. The second (background) global simulation (G-Bkgd) used natural emissions only without any anthropogenic emissions. However, the methane concentration for G-Bkgd was the same as for G-Base, ~1750 ppb (2007 data), and thus represents the current, not pre-industrial level.

Both GEOS-Chem simulations were driven by off-line meteorological fields generated by the GEOS-5 general circulation
model with 2° latitude x 2.5° longitude horizontal grid resolution and 72 vertical layers (GEOS-5, 2013). Simulations ran from the beginning of 2009 to the end of 2010 to match the North American regional modeling period (2010). (Year 2009 was the model spin-up period.) Table S1 describes the model configuration further, and Tables S2, S3, and S4 give more detail on the global anthropogenic, regional anthropogenic, and natural emission inventories, respectively.

### 2.2 CAMx simulations

The CAMx version 6.30 (Ramboll-Environ, 2016), was used to simulate $O_3$ over a North American modeling domain which covers the continental US and parts of Canada and Mexico with a 12km horizontal grid (Fig. 1) and 26 vertical layers. The




top boundary is defined by a fixed pressure level, approximately 14–17 km above ground level, which is generally in the lower stratosphere (Seidel and Randel, 2006). The simulation period covered the $O_3$ season in 2010 (March to September) with 10 spin-up days. For the base-case simulation (NA-Base), which included all anthropogenic emissions, BCs were obtained from the G-Base simulation with GEOS-Chem. The background simulation (NA-Bkgd) used only natural

emissions in the entire domain and BCs from the G-Bkgd simulation. Gas-phase chemistry was represented by the Carbon Bond 2005 (CB05) chemical mechanism (Yarwood et al., 2005). Formation of particulates was not included because the focus was on $O_3$.

Anthropogenic and fire emissions were obtained from the 2010 database developed for the Air Quality Model Evaluation International Initiative (AQMEII) Phases 2 and 3 (Pouliot et al., 2015). For the US, these emissions were developed by

projecting the National Emissions Inventory (NEI) for 2008 (US EPA, 2013) to 2010. We added lightning $NO_x$ emissions using the CAMx lightning emission preprocessor (Koo et al., 2010; Morris et al., 2012). Biogenic emissions were estimated using the Model of Emissions of Gases and Aerosols from Nature (MEGAN) (Guenther et al., 2006) with updated land cover data and emission factors (Yu et al., 2015). Canadian and Mexican wildfire emissions are not included in the AQMEII database because spatial and temporal information for these emissions was unavailable (Pouliot et al., 2015). Consequently,

contributions of natural emissions in Canada and Mexico to US $O_3$ may be somewhat underestimated in our simulations. However, the absence of the Canadian/Mexican wildfire emissions should have minimal impact on our allocation of the anthropogenic $O_3$ increment to the anthropogenic Canadian/Mexican emissions. A summary of the emissions is in Table 1, divided into the contributions from the US and the remainder of the CAMx domain.

Meteorological conditions and other auxiliary model inputs are also from the AQMEII modeling database. The

meteorological fields are from a simulation for 2010 with the Weather Research and Forecasting (WRF) model (Skamarock et al., 2008) conducted by the US EPA as part of AQMEII.

### 2.3. Source apportionment by the Path-Integral Method

The PIM determines the source contributions by integrating first-order sensitivity coefficients over a range of emissions from the background case to the base case (Dunker, 2015). The equation relating the anthropogenic increment to the source

contributions is

$$c_i^{base}(\boldsymbol{x}, t; \boldsymbol{\Lambda} = 1) - c_i^{bkgd}(\boldsymbol{x}, t; \boldsymbol{\Lambda} = 0) = \sum_{m=1}^{4} \int_P \frac{\partial c_i(\boldsymbol{x}, t; \boldsymbol{\Lambda})}{\partial \lambda_m} \, d\lambda_m \qquad (1)$$

Here, $c_i^{base}$ is the concentration of species $i$ in the base simulation at location $\boldsymbol{x}$ and time t, $c_i^{bkgd}$ is the concentration in the background simulation and $\partial c_i(\boldsymbol{x}; \boldsymbol{\Lambda}))/\partial \lambda_m$ is the sensitivity of $c_i$ to the parameter $\lambda_m$. The array $\boldsymbol{\Lambda}$ contains the parameters $\lambda_m, m = 1, \dots 4$. Each $\lambda_m$ scales the difference in emissions or BCs between the base and background cases such

that when all $\lambda_m = 0$ (i.e., $\boldsymbol{\Lambda} = 0$), we have the background case and when all $\lambda_m = 1$ ($\boldsymbol{\Lambda} = 1$), the base case. P is some path from $\boldsymbol{\Lambda} = 0$ to $\boldsymbol{\Lambda} = 1$.




In the forward direction, emissions are added along the path, and the integrals accumulate the contributions to $O_3$ from the sources due to the added emissions. Viewed in reverse, the path is a scenario for reducing emissions from the base case to achieve the background case. We chose a path on which all emissions are reduced by the same factor (a synchronous or diagonal path, represented by $\lambda_m = \lambda$, $m = 1, \dots 4$). This is an unbiased approach that assumes future controls on

anthropogenic emissions will produce similar fractional reductions in different regions. A different path could be chosen for the integration, but we know of no other assumption for emission controls that has a better justification.

For this work, we divided the anthropogenic increment to US $O_3$ into the source contributions: 1) US anthropogenic emissions in the CAMx domain; 2) Canadian/Mexican anthropogenic emissions in the domain; 3) the anthropogenic component of the lateral BCs; 4) the anthropogenic component of the top BCs. The latter two sources represent pollutants

from anthropogenic emissions outside the domain that arrive through the boundaries. Together, the four sources account for the impact of all anthropogenic emissions, worldwide, on the US $O_3$. Regarding the first two sources, we separated shipping emissions within the CAMx domain into emissions inside and outside the US Exclusive Economic Zone (NOAA, 2016a), generally a 200-nautical mile limit of the coast (Fig. 1). Emissions inside the limit were assigned to the US; outside the limit, to the Canadian/Mexican category. Also, for the last two sources, the pollutants arriving through the boundaries

consist of both primary anthropogenic emissions and secondary pollutants, e.g., $O_3$, formed from the emissions. The PIM, as implemented for this study, includes the impact of all the boundary species, and the anthropogenic component of the BCs is determined by the difference in BCs between the base and background cases.

Equation (1) is an exact mathematical relationship, but in an application, the integration must be done numerically. We used a Gauss–Legendre formula with 3 points and transformed the integration variable from $\lambda$ to $r = \lambda^{0.5}$ to improve accuracy.

The sensitivities were calculated by the Decoupled Direct Method (Dunker, 1984; Dunker et al., 2002). Additional details of the PIM are in Dunker (2015) and Dunker et al. (2015). Figure 2 illustrates schematically the relationship of the PIM to the GEOS-Chem and CAMx simulations.

## 3 Results

### 3.1. Model performance for GEOS-Chem

We evaluated modeled $O_3$ from the G-Base simulation at selected surface sites outside the US: one site in Ireland, four sites in Japan, and five sites in Canada (WMO, 2016). The site in Ireland (Mace Head) is influenced by outflow transport across the Atlantic Ocean from North America. The Japanese sites experience $O_3$ export from continental Asia to the Pacific Ocean, and the five rural sites in Canada are near the northern boundary of the CAMx modeling domain. For averages over March–September using a 40 ppb threshold, GEOS-Chem underestimated $O_3$ at all the selected sites except for Kejimkujik

in Nova Scotia, which is influenced by US outflow (Table 2). With no threshold, GEOS-Chem over-predicted $O_3$ at all sites except Mace Head. Thus, GEOS-Chem has less dynamic range than observations and tends to underestimate the larger





surface $O_3$ concentrations outside the CAMx domain but overestimate the lower concentrations. With the 40 ppb threshold, the normalized mean error is 13%–28%. The error and the correlation are both generally greater with no threshold.

Vertical $O_3$ profiles were compared to ozonesonde measurements (NOAA, 2016b) at Trinidad Head, CA, Hilo, HI, Boulder, CO, Huntsville, AL, Narragansett, RI, Summit, Greenland, and the South Pole (Table S5). Figure 3 shows the comparisons

for Trinidad Head and Hilo in April and August, 2010, two sites likely to be influenced by transport of $O_3$ from Asia. Figures S1 and S2 give comparisons for the other sites. In addition to the measurement data and results for the G-Base simulation, results are also included for the G-Bkgd simulation to indicate the impact of worldwide anthropogenic emissions on $O_3$ above the surface.

At Trinidad Head, there is relatively good agreement between the G-Base $O_3$ and the ozonesonde measurements at altitudes

up to the mid-troposphere (~7 km) that are most influential to ground-level $O_3$ in the Intermountain West and Eastern US (Nopmongcol et al., 2017). However, the G-Base $O_3$ is greater than the measurements in the marine boundary layer (<1 km) in August and less than the measurements at upper altitudes in April. At Hilo, the G-Base $O_3$ is very close to the measurements in April and to the surface measurements in August but overestimates the measurements above the surface in August. At sites interior to the US (Boulder, Huntsville Narragansett), the modeled $O_3$ agrees well with the measurements in

April below 7 km but is consistently greater than the measurements by ≥20 ppb in August. The G-Base $O_3$ at Summit agrees reasonably well with the measurements below 7 km in August but underpredicts the measurements in April. Agreement at the South Pole is very good at all altitudes.

Near the top boundary of the CAMx domain, GEOS-Chem consistently overpredicts $O_3$ for mid-latitude sites (Boulder, Hilo, Huntsville, Narragansett, Trinidad Head) (Fig. S3). From 15–25 km, the predictions for the G-Base case are up to 1200 ppb

greater than measurements. To the extent that this stratospheric air is mixed downward, it would contribute to any overpredictions by GEOS-Chem and CAMx in the troposphere.

We also compared GEOS-Chem predictions to observations at CASTNet sites, which are rural locations (US EPA, 2016a). Following Fiore et al. (2014), Fig. 4 presents a comparison of monthly average MDA8 $O_3$ from the G-Base and G-Bkgd simulations to the observations at high-altitude (>1.5 km) Intermountain West sites (except CA) and at low-altitude (<1.5

25   km) sites. (Sites are shown in Fig. S4.) In March and April, the G-Base $O_3$ agrees well with observations at low-altitude sites and somewhat underestimates observations at the high-altitude sites but, in summer, significantly overestimates the observations at all altitudes. Also, the observations show a small decreasing trend from spring to summer whereas the G-Base results show a clear increasing trend. The G-Bkgd results parallel those of G-Base, approaching the observed mean MDA8 $O_3$ for the high-altitude sites in August, suggesting that the summer $O_3$ overestimation of the G-Base simulation is

largely caused by overestimating the non-anthropogenic contribution. Other studies indicate that overestimated lightning $NO_x$ emissions in the southern US (below 35° N) may explain the positive GEOS-Chem bias in summer at the Intermountain West sites (Zhang et al., 2014; Travis et al., 2016). The positive bias in summer at the low-altitude sites may be due to overestimated lightning $NO_x$ emissions and/or excessive vertical mixing (Travis et al., 2016). GEOS-Chem may also overestimate the contribution of US anthropogenic emissions to summer $O_3$, e.g. because of coarse horizontal grid resolution



or because NOₓ emissions are overestimated in EPA's 2011 National Emissions Inventory (Travis et al., 2016), but the August ozone soundings at continental US sites (Fig S1) suggest otherwise because the $O_3$ overprediction is larger in the mid-troposphere than near ground level. Seasonal averages and the 4[th] highest MDA8 $O_3$ concentrations in the North American domain for the G-Base and G-Bkgd cases are shown in Fig. S5.

The GEOS-Chem evaluation at surface and ozonesonde sites outside the US shows both over- and underprediction compared to measurements (Table 2, Figs. 3, S2), which introduces uncertainty in the BCs for CAMx. The GEOS-Chem positive bias in summer at sites within the US (Figs. 4, S1) should not directly influence our modeling with CAMx for the North American domain.

It is difficult to compare our performance with the GEOS-Chem model to that in recent studies due to differences in the
version and configuration of the model, the emissions used, the year simulated, and the observational data compared to the model results. Given these limitations, our performance is similar to what some others have obtained (Zhang et al., 2008; Emery et al., 2012; Nopmongcol, 2016). However, Zhang et al (2014), who reduced lightning NOₓ emissions, Travis et al. (2016), who reduced lightning and anthropogenic NOₓ emissions, and Yan et al. (2016), who used three two-way nested models within GEOS-Chem, obtained better performance than we did. Fiore et al. (2014) also found better performance for
GEOS-Chem in their analysis corresponding to Fig. 4.

An alternative global model is the Geophysical Fluid Dynamics Laboratory's Atmospheric Model 3 (AM3) (Donner et al., 2011). GEOS-Chem and AM3 have important differences in biogenic isoprene emissions and chemistry, lightning NOₓ and wildfire emissions, and stratosphere–to–troposphere transport (Fiore et al., 2014). Fiore et al. (2014) found that AM3 gives greater background $O_3$ at high-altitude western surface sites in spring than GEOS-Chem. This may be due to more
stratosphere–to–troposphere transport in AM3 (Lin et al., 2012) and/or more efficient mixing of free tropospheric air into the planetary boundary layer. Emery et al. (2013) found that CAMx simulations using BCs from GEOS-Chem had better performance for maximum daily average 8-hour (MDA8) $O_3$ at EPA's Clean Air Status and Trends Network (CASTNet) sites (US EPA, 2016a) in April–May than simulations using BCs from AM3; BCs from GEOS-Chem and AM3 gave similar CAMx performance for June–September. Considering this work, we chose GEOS-Chem for the present study. However,
AM3 may simulate the day-to-day variation in the stratospheric contribution to spring $O_3$ in the western US better than GEOS-Chem (Lin et al., 2012; Fiore et al., 2014).

### 3.2. Model performance for CAMx

Figure 4 also contains results for the CAMx NA-Base and NA-Bkgd simulations at CASTNet sites. The NA-Base results are closer to the mean of the observations at both the high-and low altitude sites than are the G-Base results, and the NA-Base
results are always within one standard deviation of the mean. However, the NA-Base results show essentially no trend from spring to summer whereas the measurements show a downward trend. The NA-Bkgd results parallel the NA-Base results ~20 ppb lower at the low-altitude sites and ~15 ppb lower at the high-altitude sites.





In addition to comparing CAMx predictions to observations at CASTNet sites, we evaluated performance for the sites reporting to EPA's Air Quality System (AQS), which are urban and suburban locations (US EPA, 2016b). Figure 5 presents a comparison of predicted MDA8 $O_3$ concentrations from the NA-Base case with observations at AQS and CASTNet sites, using a zero threshold for the observations. The normalized mean bias (NMB) is 4.5%–5.1% and the normalized mean error

(NME) is 17.1%–18.1%, which suggest good performance. However, there is overprediction at the lower concentrations and underprediction at the higher concentrations similar to, but more pronounced than, that in other work (Emery et al., 2012; Nopmongcol et al., 2016, 2017).

To investigate further, we compared our BCs from GEOS-Chem to those provided by the European Centre for Medium-Range Weather Forecasts (ECMWF) for the Air Quality Model Evaluation International Initiative (AQMEII) Phase 3

(Nopmongcol et al., 2017). Figure S6 compares simulations for summer with the two sets of BCs; deposition was included but chemistry and emissions were inactive. At the AQS and CASTNet sites, the GEOS-Chem BCs generally gave greater MDA8 $O_3$ than the ECMWF BCs. This suggests that the GEOS-Chem BCs contribute to the overprediction in Fig. 5 at the lower concentrations. Plots of the simulated MDA8 $O_3$ from the NA-Bkgd case vs. the observations (not shown) display overprediction for observations < 20 ppb, so overprediction at low concentrations apparently exists in the background case

also. The reasons for the underprediction at observed concentrations > 70 ppb in Fig. 5 are not clear. Some of the underprediction could be due to the absence of the Canadian/Mexican wildfire emissions. Deriving CAMx BCs from the ECMWF model rather than GEOS-Chem would have improved performance of our NA Base case but was not viable because there was no matching simulation with zero anthropogenic emissions for our NA Bkgd case.

We also determined the NMB and NME for the NA-Base simulation, focusing on the larger concentrations by using a 40-

20  ppb threshold for the MDA8 $O_3$ observations (Table S6). The NMB and NME ranged from -4.9 % to 4.3 % and 12.1% to 14.2 %, respectively, at the AQS and CASTNet sites in spring and summer, which are similar magnitude NMB and smaller NME than with the zero threshold. This performance is comparable to that obtained by Nopmongcol et al. (2017) for AQMEII Phase 3 (Table S6) using a similar CAMx configuration and inputs (except for BCs, dry deposition scheme, lightning $NO_x$ and biogenic emissions), indicating that factors shared by these two simulations (e.g., anthropogenic

emissions, chemistry scheme, meteorology) are most influential on the larger $O_3$ concentrations.

### 3.3. Boundary concentrations from GEOS-Chem

The monthly average $O_3$ concentrations on the lateral boundaries of the CAMx domain are shown in Fig. 6 for the surface layer and layer 23, which is centered near 10 km altitude. Mostly, layer 23 represents the upper troposphere except near the northern boundary in late spring, where the tropopause is lower (Seidel and Randel, 2006) and layer 23 represents the lower

stratosphere. In the surface layer for the west, east, and south boundaries, the $O_3$ concentration is 30-45 ppb for the base case, and there is a minimum in June or July. On the north boundary, the concentration is generally lower by up to 9 ppb than on the other boundaries. For layer 23, there is a decreasing trend from spring to summer on the north, west, and east boundaries and no clear trend on the south boundary; the $O_3$ concentration is greatest on the north, similar on the west and


east and least on the south boundary. In April, layer 23 of the west boundary shows a springtime maximum in upper tropospheric $O_3$, which has been associated with events of high ground level $O_3$ in the Intermountain West (Zhang et al., 2014) when air descends from high altitude to ground level with little dilution. In the base case, the concentration for layer 23 is 20–134 ppb greater than for the surface layer, reflecting destruction at the surface. These trends in the base-case BCs

are consistent with our understanding of tropospheric $O_3$ in the Northern Hemisphere.

The boundary concentrations for the NA-Bkgd case closely parallel those for the NA-Base case on all lateral boundaries in layer 23 and on the west in the surface layer. The anthropogenic increment to the lateral boundary concentrations is 7–22 ppb in the surface layer and 8–20 ppb in layer 23. This increment is a large fraction (generally 30–60%) of the base-case boundary concentrations for the surface layer but a moderate fraction (10–21%) for layer 23. The April and August average

$O_3$ concentrations on the top boundary of the CAMx domain, at 14–17 km, are shown in Fig. S7 for the base case along with the anthropogenic increment. The increment for the top boundary is of similar magnitude to that on the lateral boundaries for layer 23 and the surface layer. However, the anthropogenic increment on the top boundary is only a small fraction (typically 1–5%) of the top boundary concentration for the base case. Thus, GEOS-Chem predicts a positive but only small influence of anthropogenic emissions on $O_3$ in the lower stratosphere, and the $O_3$ difference on the top boundary between the

NA-Base and NA-Bkgd simulations should give only a small contribution to the source apportionment (confirmed below).

### 3.4. Base-case and background ozone

The spring (March–May) and summer (June–August) seasonal averages of MDA8 $O_3$ from the NA-base and NA-Bkgd simulations are given in Fig. 7. Also shown is the average of the 10 largest MDA8 $O_3$ concentrations in the base case (T10Base) over March–September and the average over the same days in the background simulation. The T10Base dates

can differ by grid cell. For the base case in spring, the larger $O_3$ concentrations are in an arc from British Columbia through the Rocky Mountains into the Sierra Madre mountains of Mexico, across the Gulf of Mexico and along the US east coast. The background $O_3$ in spring has a similar spatial pattern with the largest concentrations of 40–50 ppb in Colorado, New Mexico, Arizona, and Mexico. However, in the base case, the $O_3$ levels in the western and eastern US are similar, but in the background case the levels are distinctly lower in the eastern than western US.

The summer average for the base case shows MDA8 $O_3$ greater than 50 ppb across most of the US. The largest concentrations are near or downwind of urban areas and in Colorado with the maximum of 69 ppb near Washington, DC. The spatial pattern of the background concentration in summer is like that in spring except that concentrations are smaller in Mexico and larger in Canada than in spring. The largest background concentrations are in Colorado and California, and the maximum is 47 ppb in the Sierra Nevada mountains.

The T10Base average for the base case shows an even more pronounced impact of urban areas with larger concentrations also in Colorado, New Mexico, and Mexico. The maximum concentration is 96 ppb, again near Washington, DC. The T10Base average for the background case assigns the largest concentrations to central Canada, the western US, and Mexico.



The maximum background of 64 ppb is in the Sierra Madre Occidental mountains of Mexico, but there are also locations in Colorado, New Mexico, Arizona, and California that exceed 60 ppb.

Table 3 gives the T10Base concentrations for the base and background cases at 10 urban and two CASTNet sites. In choosing sites to represent urban areas, we selected the site with highest design value in each urban area. (Site IDs are in Table S7.) Also in the table are the base and background concentrations averaged over the 10 days with the largest background concentrations (T10Bkgd). For the base case, the T10Base averages exceed 70 ppb at all the sites except Sacramento, Salt Lake, Big Bend, and Perkinstown, and the T10Bkgd averages exceed 60 ppb at half the sites. The background concentrations are largest at the higher elevation (> 1 km) sites: Denver, Salt Lake, and Big Bend. The T10Bkgd average for the background case ranges from 35 to 54 ppb at the 12 sites, and the T10Base average background ranges from 24 to 50 ppb. For Denver, the background concentration is 69% and 79% of the base-case concentration with the T10Base and T10Bkgd averages, respectively.

### 3.5. Source apportionment of the anthropogenic increment

The PIM quantifies source contributions by numerically integrating Eq. (1), and we evaluated the accuracy of the numerical integration by comparing the sum of the contributions to the anthropogenic increment (right-hand side vs. left-hand side of Eq. (1), respectively). Including all surface grid squares in the CAMx domain, the sum of the contributions to MDA8 $O_3$ correlated closely ($R^2 = 0.999$, least squares slope = 0.99) with the anthropogenic increment in March and June (Fig. S8). At selected AQS and CASTNet sites, the maximum error and average error over the seven-month simulation are <3.5 ppb and < 1.5 ppb, respectively (Fig. S9). Errors are smaller at the CASTNet sites than at the AQS sites, most likely because the concentrations are smaller at the CASTNet sites. This accuracy is very similar to that in our previous application of the PIM for a 3D simulation. (Dunker et al., 2015).

The anthropogenic increment to MDA8 $O_3$ in ppb, based on the T10Base average, is shown in Fig. 8 along with the source contributions to the increment from the PIM. Analogous plots for the spring- and summer- average MDA8 $O_3$ are in Figs. S10 and S11, respectively. The relative source contributions in percent are in Figs. 9, S12, and S13. These figures do not show the contribution from the anthropogenic component of the top BCs because this contribution to surface concentrations is always very small, ≤ 0.5 ppb or ≤ 3% of the increment. The small magnitude is consistent with the small fraction of the top BCs arising from anthropogenic emissions, as predicted by GEOS-Chem (Fig. S7). The small contribution of anthropogenic top BCs does not preclude a larger contribution from natural stratospheric $O_3$. We omit further analysis of the anthropogenic top BCs contribution.

The anthropogenic increment is larger to the east of mid-Texas and along the west coast than in the states in between and generally larger in summer than spring. The maximum increment is 30 ppb in spring along the Louisiana coast, 42 ppb in summer near Washington, DC, and 75 ppb for the T10Base average, also near Washington, DC. The US anthropogenic emissions are the largest contributor in ppb and percent to the anthropogenic increments in the eastern US and California. This is apparent in Figs. 8, 9 and also in the source contributions for the T10Base average in Table 3. For the cities in Table





3 in the eastern US and California (all but Denver and Salt Lake), the US anthropogenic emissions contribute 39–53 ppb (86–93%) of the T10Base increments.

The anthropogenic lateral BCs are the second most important contributor in ppb to the anthropogenic increment at most locations in the domain (Fig. 8). The contributions are largest for higher elevation US sites in the Intermountain West and

5 sites closest to the boundaries. The lateral BCs contribute 8–9 ppb (39–51%) of the T10Base anthropogenic increments at Denver, Salt Lake, and Big Bend (Table 3). At Perkinstown (northern WI), the lateral BCs contribute 6 ppb (30%) of the increment. The contribution of the lateral BCs is largest near the west boundary of the CAMx domain, decreases across the US, and increases from the US east coast toward the east boundary. The increase toward the east boundary is greatest in spring in the northeast and may be due to transport of Canadian emissions and $O_3$ outside the CAMx domain and re-

10 circulation back through the east boundary via the BCs from GEOS-Chem. On a relative basis, the lateral BCs are important where the US emissions are not very important and vice versa (Figs. 9, S12, S13).

The Canadian/Mexican anthropogenic emissions are third in importance, affecting the northern, east-coast, and southwest US and some interior states. In summer, these emissions also contribute ~2 ppb to the anthropogenic increment along the west coast, south to San Francisco (Fig. S11). The spatial pattern is consistent with prevailing summer surface winds, which

circulate anti-cyclonically around the Eastern Pacific subtropical high pressure area. At sites in Table 3 close to Canada and Mexico, Cleveland and Big Bend, the Canadian/Mexican emissions contribute 3 ppb (7%) and 2 ppb (12%), respectively, to the T10Base anthropogenic increment.

Table 3 also contains the source contributions obtained with the T10Bkgd average. The anthropogenic increment from the T10Bkgd average is 6–37 ppb lower at the urban sites and 4–8 ppb lower at the CASTNet sites compared to the increment

from the T10Base average. This reduction is due to a reduced contribution from the US emissions. The contribution of the Canadian/Mexican emissions remains about the same at all the sites with the T10Bkgd average, and the contribution of the lateral BCs increases by 2–5 ppb at the lower elevation urban sites. The net effect is that the relative importance of the lateral BCs is significantly increased for the days with the largest background concentrations. For five sites, the contribution exceeds 50% of the anthropogenic increment and at Denver and Big Bend, the lateral BCs account for 71% and 67% of the

increment, respectively.

Figure 10 has time series of the base-case and background concentrations and the source contributions for five of the sites in Table 3. Time-series plots for other AQS and CASTNet sites are in Figs. S14, S15, respectively. The background concentration is 20–40 ppb at Cleveland and Dallas but generally greater, 30–50 ppb, at Denver and Big Bend and a larger fraction of the base-case concentration at the latter sites. At Perkinstown, the background is 20–30 ppb in spring but greater

in July and August, including a peak of 58 ppb on July 6. The contribution from the lateral BCs is generally larger in spring (5–15 ppb) than in summer (≤ 5 ppb). At Cleveland, Dallas, and Perkinstown, the larger MDA8 $O_3$ concentrations in the base case are driven by the US emissions (except for July 6 at Perkinstown). The US emissions contribute to the base-case concentrations at Denver and Big Bend during summer, but are less important (< 20 ppb) than at other sites. The



contribution of Canadian/Mexican emissions is largest for Cleveland and Big Bend, with maximum contributions at these sites of 22 ppb and 13 ppb, respectively.

There are negative contributions from US anthropogenic emissions on some days in Cleveland, Dallas, and Denver. These are situations in which large $NO_x$ emissions in the base case inhibit $O_3$ formation. The sensitivity of $O_3$ to the emissions is

positive at the starting point of the integral in Eq. (1) (background case) but negative at the ending point (base case), and if the emissions are large enough, the total integral is negative. The negative contributions, which are small in magnitude, merely indicate that large reductions in the US anthropogenic emissions are necessary on these days before $O_3$ decreases in response.

## 4 Conclusions

We used the CAMx regional model in a one-way nest within the GEOS-Chem global model to predict North American $O_3$ concentrations in March–September 2010 for a base case with all emissions present and a natural-background case without worldwide anthropogenic emissions. The difference between these two simulations, the anthropogenic increment, was allocated to the anthropogenic sources: US emissions, Canadian/Mexican emissions, and the anthropogenic components of the lateral BCs and the top BCs. The PIM was used for this source allocation, which required sensitivities from three

additional simulations with emissions and BCs intermediate between the base and background cases. The major unique features of the study are allocating the anthropogenic $O_3$ increment, rather than the total concentration, and estimating contributions of the anthropogenic components of the BCs, rather than the total BCs (Lefohn et al., 2014; Dolwick et al., 2015; Baker et al., 2015; Nopmongcol et al., 2017).

The predicted natural background MDA8 $O_3$ is larger in the western US and Mexico than in the eastern US. The spatial

pattern is similar in spring and summer with the exceptions that concentrations are smaller in Mexico and larger in Canada in summer. The largest background MDA8 $O_3$ in the US is in the mountainous areas of Colorado, New Mexico, Arizona, and California. For the larger MDA8 $O_3$ concentrations in the base case (T10Base average), the background $O_3$ at Denver is 50 ppb, which is 69% of the corresponding base-case concentration (72 ppb). The background $O_3$ exceeds 60 ppb in some other western US locations.

Using the T10Base average, the US anthropogenic emissions are the largest contributor in ppb and percent to the anthropogenic $O_3$ increments in the eastern US and California. Second in importance are the contributions of the anthropogenic lateral BCs, which are largest for the higher elevation US sites in the Intermountain West and sites closest to the boundaries. The Canadian/Mexican emissions are third in importance, affecting the northern, east-coast, and southwest US and some interior states. The contribution of the anthropogenic top BCs is always very small.

We also examined results for the larger MDA8 $O_3$ in the background case (T10Bkgd average). The anthropogenic $O_3$ increment is smaller with the T10Bkgd average than the T10Base average due to a reduced contribution from US emissions. The contribution of the Canadian/Mexican emissions remains about the same, and the contribution from the lateral BCs



increases by up to 5 ppb at lower elevation urban sites. The net effect is that the relative importance of the lateral BCs is significantly increased for the days with the largest background concentrations, up to about 70%.

GEOS-Chem may have less stratosphere-troposphere exchange than AM3, which may cause the smaller background $O_3$ concentrations with GEOS-Chem than AM3 found by Fiore et al. (2014) for the western US in spring. Hence, using BCs from AM3 for the CAMx simulations may give greater background and base-case $O_3$ at western sites in spring, which could affect the T10Bkgd results in Table 3. The T10Base results would not be affected because all the days included in the average are in summer except for the Big Bend average, which includes some spring days.

Global and regional models are continuing to evolve as new data and analyses become available. Estimates of lightning $NO_x$ emissions have been reduced in recent studies with GEOS-Chem (Zhang et al., 2014; Travis et al., 2016) from those in the version of GEOS-Chem we used (Zhang et al., 2011). Using satellite data, Pickering et al. (2016) estimated even lower lightning $NO_x$ emissions (per flash) over the Gulf of Mexico than Zhang et al. (2014) and Travis et al. (2016). Such changes would affect the BCs obtained from GEOS-Chem for the south boundary of our CAMx domain but also the lightning $NO_x$ emissions used in CAMx.

Several recent studies have concluded that the 2011 NEI overestimates US anthropogenic $NO_x$ emissions (Anderson et al., 2014; Goldberg et al., 2016; Souri et al., 2016; Travis et al., 2016). Our CAMx simulations used the 2008 NEI projected to 2010, but the procedures used in developing different versions of the NEI are similar enough that any overestimation of $NO_x$ emissions in the 2011 NEI likely implies overestimation in the 2008 NEI as well. Any overestimation in our inventory would directly cause an overestimate of the source contribution for US anthropogenic emissions. Our GEOS-Chem simulation did not include the changes of Travis et al. (2016), so overestimation in the inventory would also indirectly affect the source contributions via the BCs to the extent that there is recirculation of pollutants from inside the North American domain to the outside and then back.

Other recent work has suggested changes to GEOS-Chem for $O_3$ generation in wildfire plumes (Zhang et al., 2014; Lu et al., 2016), vertical mixing in the lower troposphere (Travis et al., 2016), and the chemistry (Fisher et al., 2016; Schmidt et al., 2016; Sherwen et al., 2016). Of note, adding halogen chemistry decreases the global tropospheric $O_3$ burden in GEOS-Chem by 14%–19% (Schmidt et al., 2016; Sherwen et al., 2016). As some of these proposed modifications are implemented in global and regional models, predictions of the background $O_3$, the anthropogenic increment, and the source contributions to the increment are likely to change.

**Supplementary information**

Supplementary information associated with this article is found in the online version at doi:xxx.



## Competing interests

The authors declare that they have no conflict of interest.

## Acknowledgments

We thank the Atmospheric Impacts Committee of the Coordinating Research Council for supporting this work and Prof.
Daniel Jacob, Harvard University, for helpful discussions.

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

**Table 1. Average daily emissions in the CAMx modeling domain for March–September 2010.**

| Source | US emissions (tons/day) | | | Canadian/Mexican emissions (tons/day) | | |
|---|---|---|---|---|---|---|
| | NO$_x$ | VOC | CO | NO$_x$ | VOC | CO |
| Anthropogenic (inland) | 41,206 | 31,194 | 139,750 | 7,377 | 7,596 | 26,874 |
| Shipping | 1,927 | 61 | 163 | 1,290 | 40 | 106 |
| Fires | 559 | 6,054 | 37,851 | | | |
| Lightning | 5,983 | | | 4,577 | | |
| Biogenic | 1,634 | 125,249 | 18,092 | 717 | 47,471 | 4,810 |
| Total | 51,308 | 162,557 | 195,855 | 13,961 | 55,107 | 31,790 |





**Table 2. Model performance metrics for hourly O₃ from the GEOS-Chem base case for the March–September, 2010 period with zero or 40 ppb thresholds for the observed concentrations.**

| Site | Latitude/ Longitude | Threshold (ppb) | NMB[a] (%) | NME[a] (%) | RMSE[a] (ppb) | R[a] |
|---|---|---|---|---|---|---|
| Mace Head | 53.33/-9.90 | none | -4.4 | 13.1 | 6.0 | 0.68 |
| Ireland | | 40. | -15.2 | 15.7 | 8.2 | 0.25 |
| Tsukuba | 36.05/140.13 | none | 20.9 | 42.1 | 15.3 | 0.73 |
| Japan | | 40. | -13.8 | 20.0 | 14.7 | 0.60 |
| Ryori | 39.03/141.82 | none | 4.5 | 23.5 | 12.2 | 0.38 |
| Japan | | 40. | -12.3 | 15.5 | 10.3 | 0.26 |
| Minamitorishima | 24.28/153.98 | none | 20.8 | 37.7 | 10.0 | 0.66 |
| Japan | | 40. | -28.1 | 28.1 | 16.5 | 0.28 |
| Yonagunijima | 24.47/123.02 | none | 7.5 | 29.5 | 11.3 | 0.84 |
| Japan | | 40. | -15.4 | 16.7 | 11.2 | 0.52 |
| Algoma | 47.03/-84.38 | none | 46.1 | 57.6 | 19.5 | 0.20 |
| Canada | | 40. | -14.7 | 17.3 | 9.0 | 0.07 |
| Bratt's Lake | 50.20/-104.71 | none | 33.5 | 44.8 | 15.4 | 0.39 |
| Canada | | 40. | -5.0 | 20.5 | 11.8 | 0.16 |
| Chapais | 49.82/-74.98 | none | 26.4 | 36.9 | 13.7 | 0.48 |
| Canada | | 40. | -8.5 | 13.3 | 7.3 | 0.46 |
| Experimental Lakes | 49.67/-93.72 | none | 33.2 | 40.6 | 15.5 | 0.27 |
| Canada | | 40. | -4.4 | 14.9 | 8.7 | 0.13 |
| Kejimkujik | 44.43/-65.20 | none | 37.7 | 42.8 | 16.1 | 0.35 |
| Canada | | 40. | 1.0 | 13.8 | 8.4 | 0.39 |

[a] NMB =normalized mean bias; NME = normalized mean error; RMSE = root mean square error; R = correlation coefficient





**Table 3. Contributions of anthropogenic sources to the anthropogenic increment of MDA8 O$_3$ at AQS and CASTNet sites.[a]**

| Site[d] | T10Base (ppb)[b] | | | | | T10Bkgd (ppb)[c] | | | | |
|---|---|---|---|---|---|---|---|---|---|---|
| | Base | Bkgd | US anthro | Can/Mex anthro | Anthro lateral BCs | Base | Bkgd | US anthro | Can/Mex anthro | Anthro lateral BCs |
| Atlanta, GA | 83.3 | 28.4 | 48.3 | 0.4 | 3.6 | 60.6 | 41.3 | 11.3 | 0.4 | 6.3 |
| Boston, MA | 75.5 | 24.4 | 45.5 | 1.8 | 2.8 | 55.6 | 35.1 | 13.2 | 1.9 | 5.2 |
| Cleveland, OH | 73.1 | 26.1 | 39.8 | 3.2 | 3.4 | 58.1 | 40.8 | 9.7 | 1.9 | 5.7 |
| Dallas, TX | 82.5 | 31.3 | 45.3 | 0.6 | 3.5 | 57.1 | 42.6 | 5.2 | 0.4 | 7.8 |
| Denver, CO | 72.2 | 49.9 | 11.9 | 0.2 | 9.4 | 68.2 | 54.1 | 3.4 | 0.5 | 9.6 |
| Pittsburgh, PA | 76.2 | 25.5 | 46.0 | 0.8 | 3.3 | 62.7 | 42.9 | 12.1 | 0.9 | 6.8 |
| Sacramento, CA | 69.9 | 24.9 | 39.3 | 0.3 | 4.6 | 60.3 | 37.3 | 12.9 | 0.4 | 9.5 |
| Salt Lake, UT | 66.2 | 43.7 | 13.2 | 0.3 | 8.4 | 62.4 | 46.2 | 6.6 | 0.3 | 8.9 |
| St. Louis, MO | 76.3 | 26.1 | 45.8 | 0.5 | 3.0 | 55.6 | 40.1 | 6.9 | 0.7 | 8.0 |
| Washington, D.C. | 83.8 | 25.5 | 53.4 | 1.2 | 2.8 | 59.8 | 38.6 | 13.2 | 0.7 | 7.0 |
| Big Bend, TX | 62.7 | 46.4 | 6.0 | 2.0 | 8.3 | 60.7 | 48.6 | 2.7 | 1.2 | 8.2 |
| Perkinstown, WI | 57.6 | 38.3 | 13.0 | 0.6 | 5.8 | 54.7 | 43.0 | 5.2 | 1.1 | 5.5 |

[a] Anthropogenic component of the top BCs contributes ≤ 0.1 ppb.

[b] Average of the 10 days with largest MDA8 O$_3$ in the base case. Days for the background case are the same as the base case.

[c] Average of the 10 days with largest MDA8 O$_3$ in the background case. Days for the base case are the same as the background case.

5   [d] IDs are in Table S7.





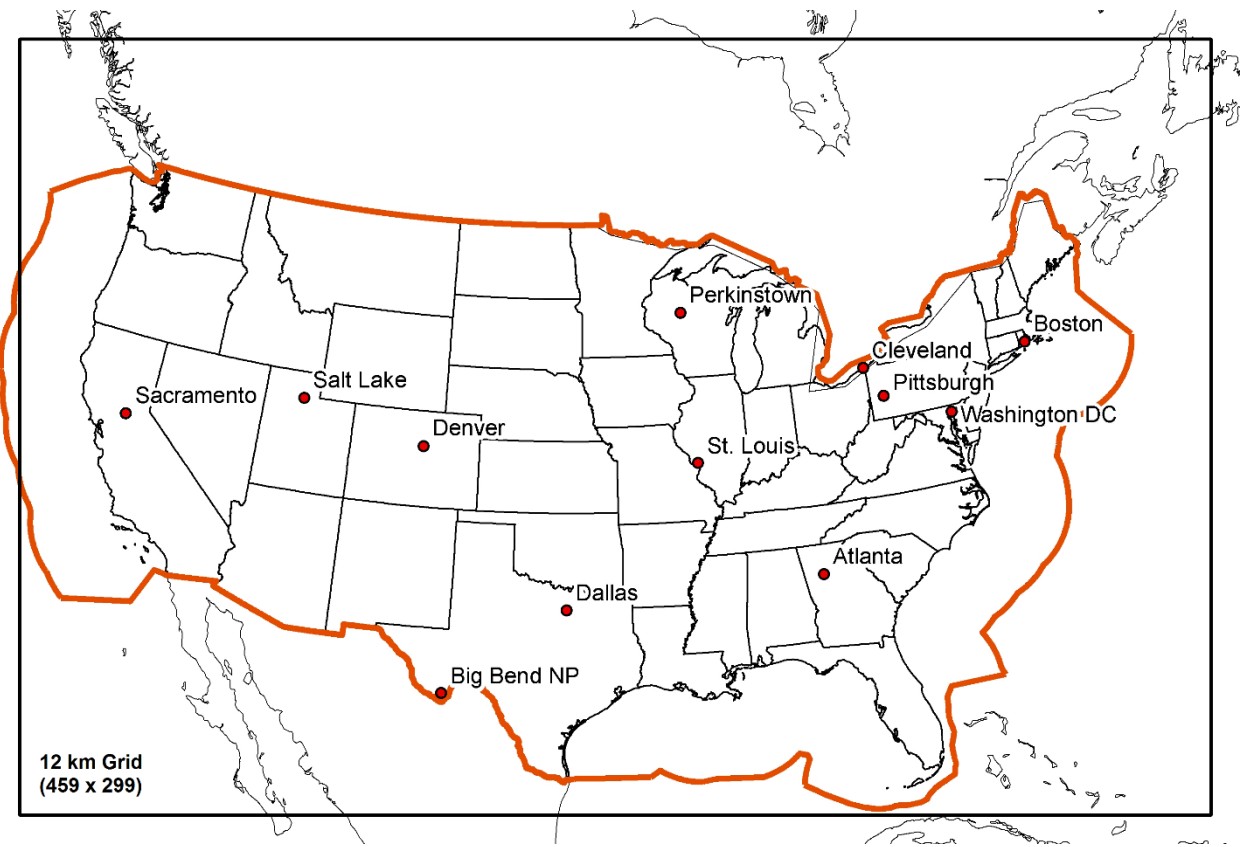

**Figure 1. The modeling domain for CAMx showing monitoring sites included in the analysis. The orange line is the US Exclusive Economic Zone. Shipping emissions inside the zone were assigned to the US and outside the zone combined with Canadian and Mexican anthropogenic emissions.**



# Attributing Ozone by the PIM

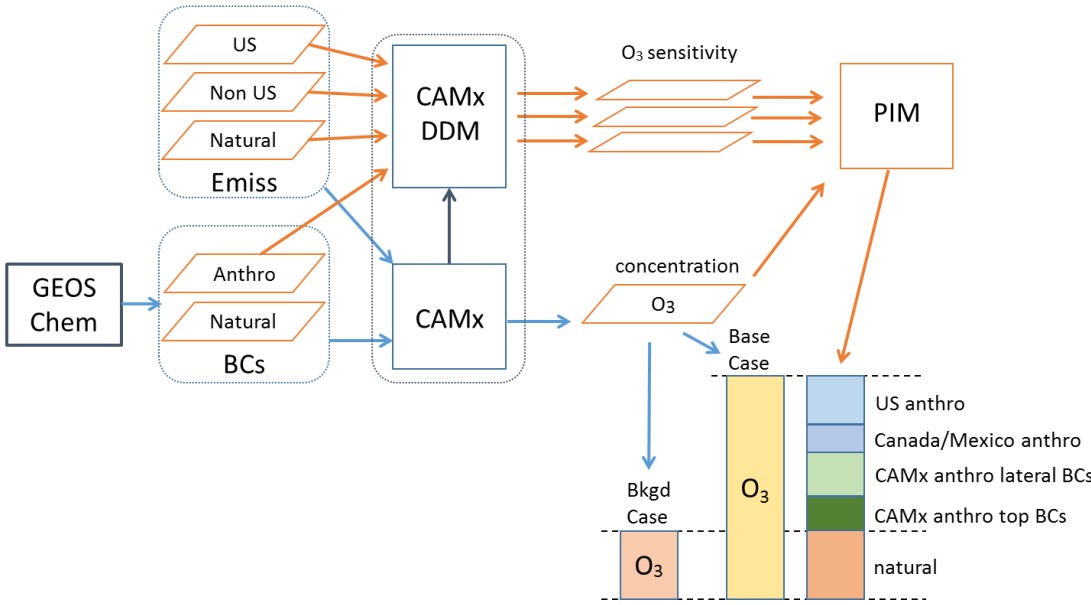

**Figure 2. Schematic diagram of source apportionment by the path-integral method (PIM). Two worldwide GEOS-Chem simulations with and without anthropogenic emissions provide boundary concentrations (BCs) for the corresponding CAMx simulations. Two CAMx simulations with and without anthropogenic emissions give the base and background cases for North America. Three CAMx simulations with emissions and BCs between the base and background cases, using the decoupled direct method (DDM), provide the sensitivities needed to allocate the anthropogenic increment to O₃ (base minus background cases) to four source categories using Eq. (1).**





**Figure 3. Comparison of GEOS-Chem vertical O₃ profiles for the base and background cases in April and August to ozonesonde measurements at a site on the west of the continental US (Trinidad Head) and west of the CAMx domain (Hilo). Monthly averages are shown.**



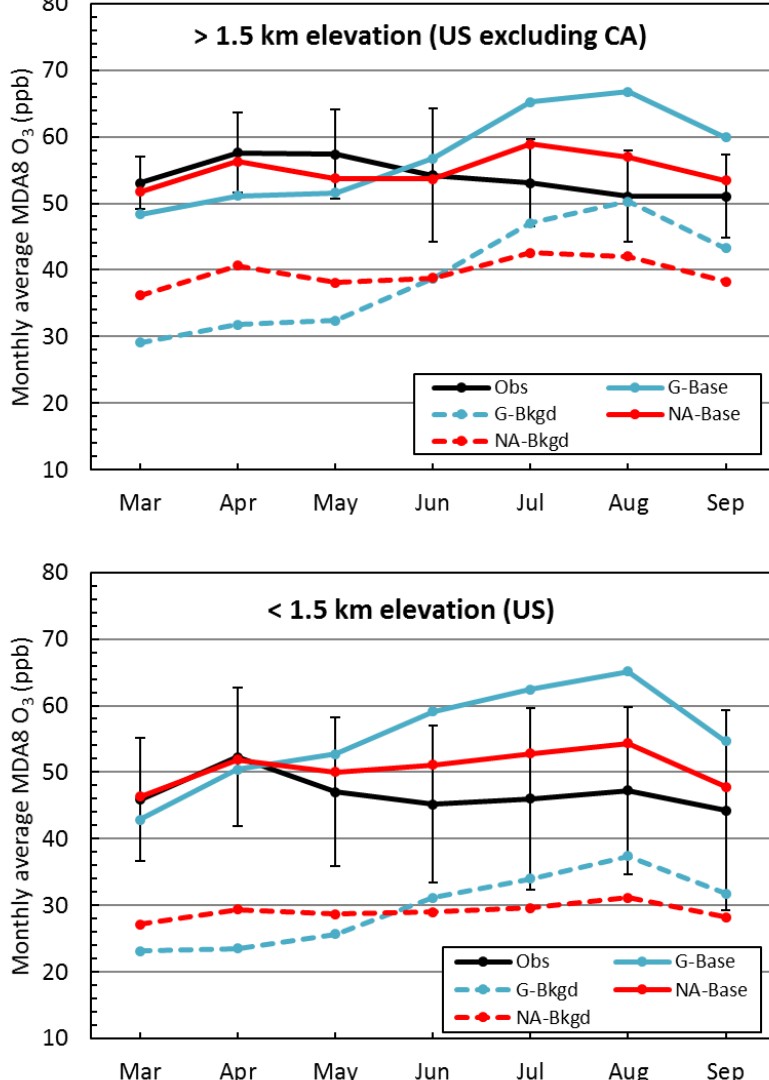

**Figure 4. Observed and modeled monthly average MDA8 O$_3$ at high-and low-altitude CASTNet sites in 2010. G-Base and G-Bkgd are the GEOS-Chem global model base and background cases whereas NA cases are the CAMx North American model. Error bars for observations indicate one standard deviation. Site locations are shown in Fig. S4.**



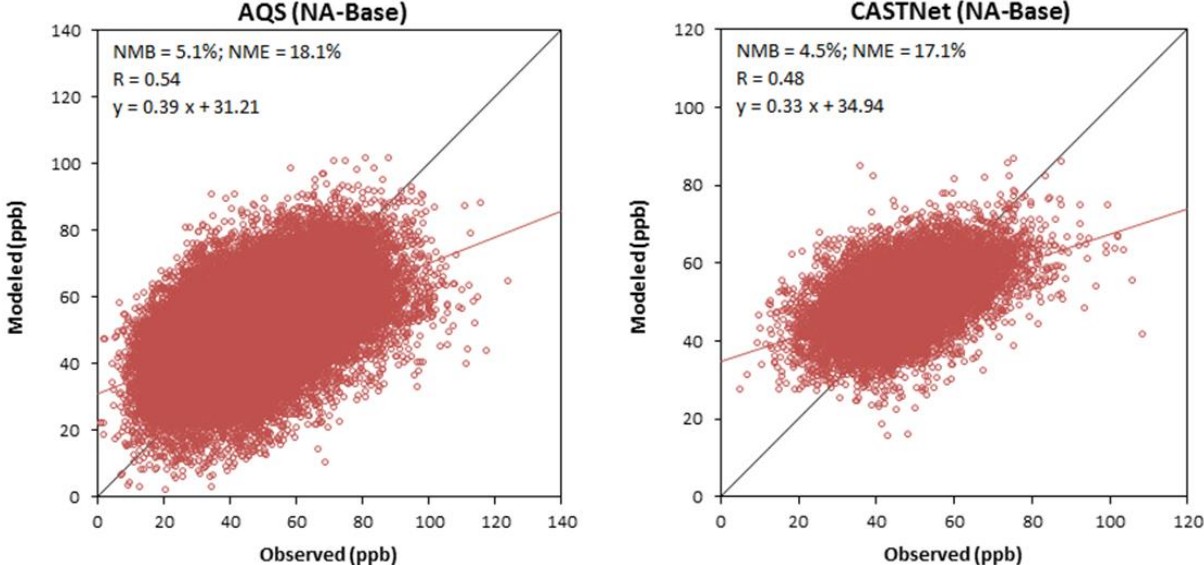

**Figure 5. Scatter plots of modeled MDA8 O₃ concentrations from the CAMx base case vs. observations at AQS and CASTNet sites for March–September, 2010. The black and red lines are one-to-one and linear regression lines, respectively. Normalized mean bias (NMB) and normalized mean error (NME) were calculated with no threshold.**



**Figure 6. Monthly average lateral boundary O₃ concentrations for the CAMx surface layer and the 23rd layer, as provided by the GEOS-Chem simulations.**



**Figure 7. MDA8 O₃ concentrations from the CAMx base (a–c) and background (d–f) simulations averaged over spring (March–May), summer (June–August), and the 10 days in the base case with the largest concentrations (March–September, 2010).**




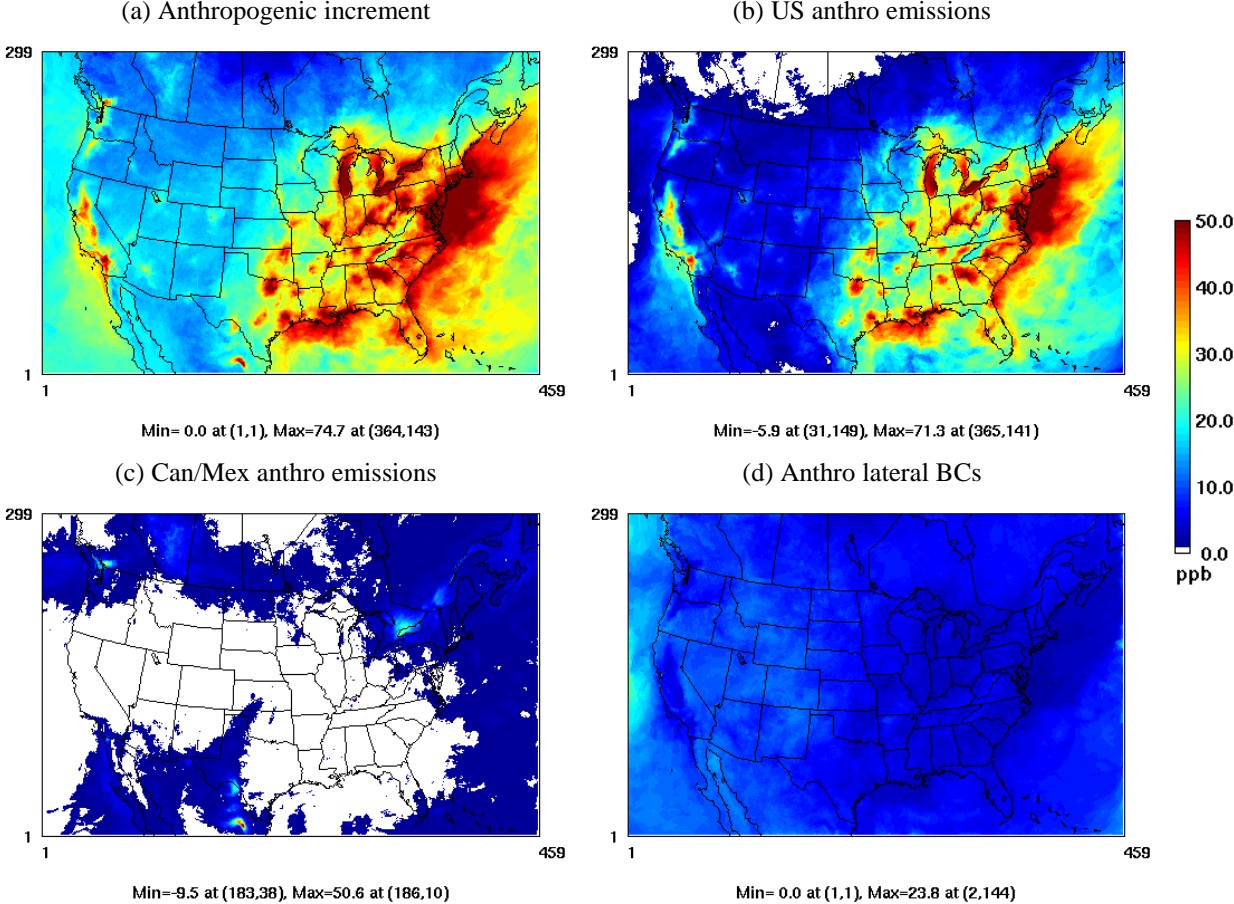

**Figure 8.** **The anthropogenic increment (a) for the average of the top 10 MDA8 O₃ concentrations in the base case (T10Base) and the contributions (b–d) to this increment. The contribution from the anthropogenic component of the top BCs is ≤ 0.5 ppb.**



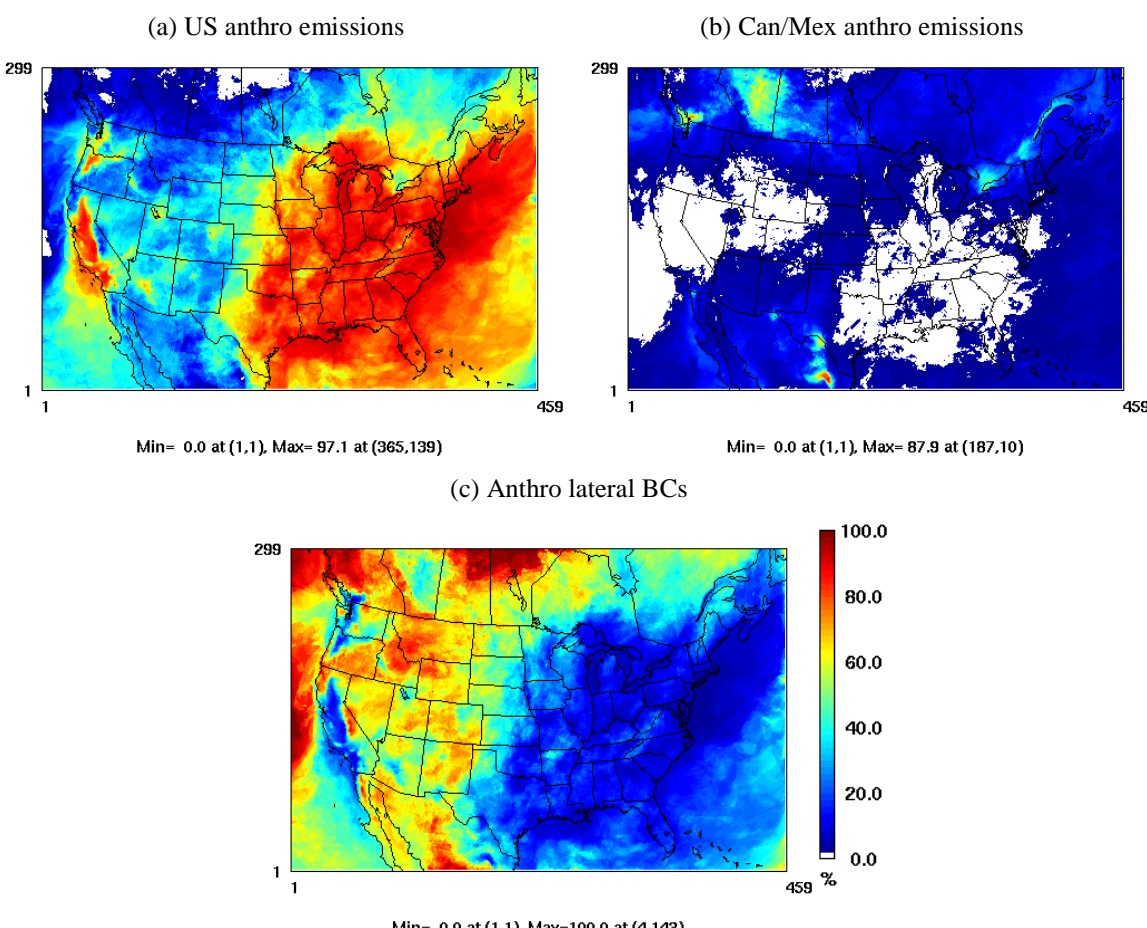

**Figure 9. Relative contributions in percent to the anthropogenic increment of the top 10 MDA8 O₃ concentrations in the base case. The contribution from the anthropogenic component of the top BCs is ≤ 3%.**



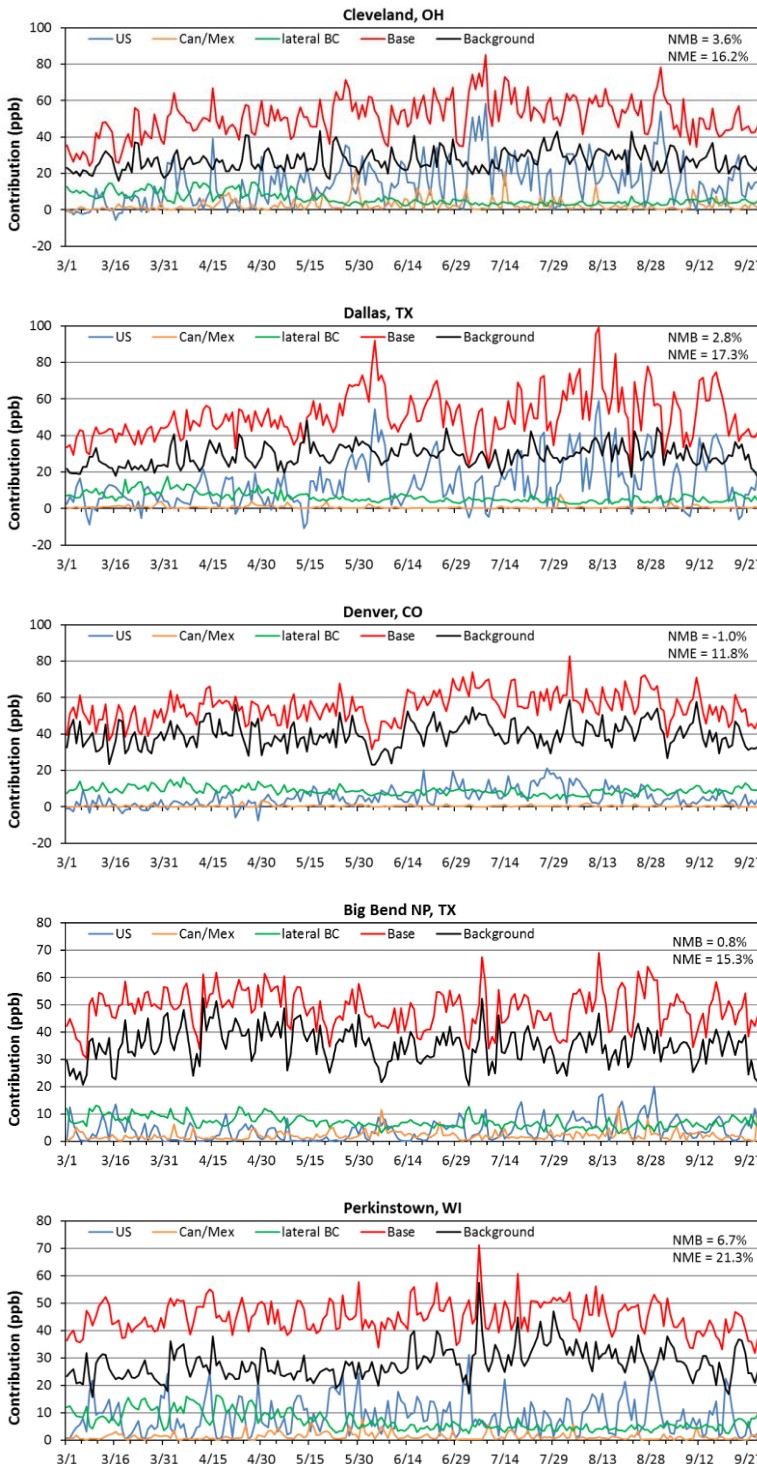

**Figure 10. Anthropogenic contributions to MDA8 O₃ at selected AQS and CASTNet sites along with base-case and background concentrations and model performance statistics (no threshold).**