# Peer review of "Contributions of foreign, domestic and natural emissions to US ozone estimated using the path-integral method in CAMx nested within GEOS-Chem"

_Atmospheric Chemistry and Physics, 2017_

## Referee Comment (RC1) · Anonymous Referee #1 · 21 Jun 2017

This manuscript applies a relatively new technique, the path-integral method, to characterize contributions to ozone over the United States. It applies two widely used models, CAMx nested within GEOS-Chem, to conduct its analysis. The main focus is on characterizing background and base ozone conditions, which have received heightened interest as the U.S. has moved to tighter ozone standards.

The manuscript is clearly written and its methods are sound. Model performance and results are in line with previous studies.

[Figure]

What is relatively novel here is the use of the path-integral method. As such, additional explanation and illustration should be provided in the final paper. Specifically, in Figure 2, it would be helpful to illustrate what happens in the "PIM" box. Also, a bit more explanation would be helpful regarding Equation 1. In particular, it is unclear to me what is happening along the 4 steps – is each source being reduced 1/4 at a time?

Moreover, it is unclear to me that PIM has a "unique capability to allocate the difference in O3 between two simulations..." (p. 2, lines 27-28). Don't zero-out and tracer methods also do so? Given the nonlinearities of ozone response to emissions, there is no uniquely correct answer to apportionment. Better clarity on how to interpret PIM relative to other methods would be useful.

Soil NOx, lightning NOx, biogenic VOC, and wildfire emissions are all very important to background O3. It should be clarified how soil NOx was estimated, and whether each of these sources was modeled consistently between GEOS-Chem and CAMx.

Finally, the authors could consider discussing the implications of their findings for the setting of ambient standards for O3, which is an urgent topic in the US now. That so many sites have base T10 concentrations above 60 ppb suggests certain limits under consideration may be unattainable in some regions. US NAAQS are formulated based on 4th highest O3 at the most polluted monitor in a region. Given that this study looks at T10 rather than 4th highest, at a single monitor rather than worst, with models that tend to underpredict peak O3, and with the impossibility of removing all anthropogenic emissions globally all suggests attainment of NAAQS below 70 ppb may be unrealistic.

---

## Referee Comment (RC2) · Anonymous Referee #2 · 27 Jun 2017

The manuscript by Dunker et al. estimates source contributions to the anthropogenic increment of O3 in the US. A one-way nested modeling framework combining GEOS-Chem and CAMx is used, allowing the authors to address global anthropogenic vs. background contributions.  A relatively novel source attribution method, the path-integral approach, is used to alleviate some drawbacks of more traditional (brute-force, tracer) methods).  The authors evaluate model performance as well as source attributions.  The findings are useful, timely, and generally well explained and examined. Model performance is typical for these types of tools, and the authors explore reasons

for discrepancies as well as alternative global modeling values. While some more work could be done to propagate this level of uncertainty into the final source attribution estimates, the overall technical approach is adequate. The introduction is a bit rough and needs some more work, the abstract is too vague, more rigorous and careful discussion and analysis is required to highlight the benefits of PIM, and there are a few other clarifying questions throughout on minor details. I believe in total these amount to minor revisions, and expect this article will be ready for publication in ACP without need for further review.

Main comments:

1.18-26: This is a good qualitative summary, but I would appreciate a more quantitative abstract. Meanwhile, with these very general descriptions, many of the statements are very obvious to readers familiar with the issue of background O3. For example, stating that contributions to background O3 from lateral BC's is largest for sites located near the boundaries seems quite obvious, although here one might wonder if this statement truly applies to all sites or just those near boundaries with mostly inflow conditions. Other qualitative descriptions (largest, closest, larger, reduced, increases, increased,...) would significantly benefit from quantitative support. The abstract is not presently overly long and could easily be revised to contain such information.

general: The introduction is rather limited. The writing style is curt and almost short-hand at times (see a few specific comments below); references and explanations are used somewhat casually. Overall, it reads like a first draft and would benefit from a much more polished presentation befitting the extensive experience of the authors.

general: The authors perform some adequate model evaluation, and discuss reasons for different model biases. However, this sense of the magnitude of the model performance does not make its way, quantitatively, into presentation of the source attribution results. Would the authors be able to e.g. include some estimates of ranges in Table 3, or elsewhere?

[Figure]

6.2. This is an interesting and useful finding. GEOS-Chem is a widely used model. Have previous studies made similar characterizations?

6.30 - 7.4: I believe other studies have also indicated that lacking halogen chemistry (Br and Cl) can lead to high GEOS-Chem concentrations, as well as how isoprene nitrate species are recycled, or underestimation of O3 dry deposition. Later I see that halogen chemistry is mentioned in the conclusions.

Fig 7 and associated text: Indeed, background concentrations are higher than normal in an absolute sense when focusing just on days with the base case concentrations are high. But are they also higher in a relative sense? I would like to see another column to Fig 7 that shows the background values divided by the base for each row. Relative concentrations are currently only mentioned in this section for Denver, on line 10 of page 10. I can evaluate them myself for select sites using Table 3, but would appreciate more discussion be added.

general: The title and framing makes me anticipate a bit more rigorous discussion of source attribution methods than what was included in this paper. The 6 specific points below address specific questions about methodology; in general though I wonder if the authors would really like to demonstrate the advantage of PIM if they would present their findings side-by-side with those from brute-force or tracer methods, both in terms of the estimates of background concentrations and attributions and also the computational intensity (CPU and memory) of obtaining these estimates.

2.23-26: While the description of the downsides of tracer methods is technically correct, I'm not sure the extent to which these are real limitations. For example, by how much do modern tracer methods not sum to the total increment? And while all nonlinear indirect reactions may not be included, to what extent are the most important ones included, which capture e.g. >90% of the O3 formation mechanism? A more practical downside, not mentioned here but which should be, is the computational burden of these tracer methods does not scale well when many sources (regions, times) are required, as the

approach becomes memory intensive.

3.1: I feel like this statement is a bit unfair to PIM. The approach is more computationally intensive than a single brute-force sensitivity calculation. But PIM is efficient for obtaining the type of results that it is designed to calculate, compared to estimating the same contributions using other approaches. It also may be more CPU intensive whereas tracer methods would be more memory intensive. So, some more nuanced text here would be appreciated.

4.7: Particulates impact O3 via heterogeneous chemistry and photolysis. Many studies in the literature report these influences, which may be several ppb under particular conditions. Please explain further, quantitatively, and with references, why such effects are negligible in this case. My hunch is the authors will be forced to admit this decision was based, at least in part, on computational convenience, although I'm not explicitly sure why (does CAMx 6.3 not support DDM calculations for particulates? etc.).

5.4-6: There is no evidence provided to support the claim that this approach is "unbiased". It seems that in fact this assumption introduces a subjective bias in the analysis, which is the restriction that all precursor emissions change uniformly, which is clearly not representative of real-life conditions wherein emissions control measures target individual species (e.g., diesel NOx regulations). I think the rational here is one of simplicity and generality; if the study were more directed at the effects of particular control measures, different paths could be selected. One could also consider how emissions have changed, historically, and use those to define the path. But it doesn't appear that an analysis of emissions trends formed the basis of this statement, at least none is presented or cited.

5.11: Does this really account for the impact of all anthropogenic emissions on US O3? For example, if US anthropogenic NOx depleted some biogenic VOC concentrations, which then transported out of the US domain temporarily and then recirculated back into the domain at lower concentrations than would have occurred w/o these anthropogenic NOx emissions, would this be captured? I don't think so. At least this isn't mentioned on lines 5.14-15. I'm not arguing this is a substantial impact, quantitatively, just that the description of the method is neglecting some assumptions.

Section 3.4: The T10 metric is interesting, but it isn't clear to my why it was chosen. Are their findings sensitive to the use of 10 days rather than 3 or 30? Is there a policy relevance, like the 4th highest MDA8?

Minor comments / corrections:

Title: Is CAMx really "nested" within GEOS-Chem in the usual sense of a nested-grid model, or is it just using boundary conditions from GEOS-Chem (i.e., a one-way nesting)?

1.24: Not sure that a verb "increased" is appropriate here — consider "higher".

2.1: "O3 background in the absence of anthropogenic emissions" is a vague phrase that needs to be more carefully written. A formal definition of what is meant by "background O3" in this particular manuscript should be clearly defined before this term is used. Further, assuming this has been defined, the rest of this statement seems redundant, for what other O3 besides the background would be present in the absence of anthropogenic emissions?

2.3: Emissions are not transported, they are emitted. Species that are emitted are transported. Please tighten of the language in this regard here, and throughout.

2.3-4: Please provide a reference or references.

2.8: Unless the acronym "NAB" is introduced here and used later, it doesn't make sense to capitalize Background.

2.13: the phrase in parentheses is missing some words in order to be grammatically correct.

2.18: The chemistry —> Chemistry

2.19: parenthesis not needed.

3.30: Use "CMAx" or "The CAMx model". Remove extra comma.

Eq 1: This would be a more general equation if the "4" were replaced by e.g. M, where M is defined to be the number of sources. Also, it seems implicit that the integral bounds are lambda=0 to lambda=1; it's not clear why "P" is used instead.

4.28: An "array" is a computational object, not a mathematical one. Perhaps the authors mean "vector"?

5.1: This first sentence is confusing. What is the "direction" being referred to here? Some model time integration? How are "emissions added"?

Fig 1: A small point of clarification — how are shipping emissions within the US Exclusive Economic Zone but outside the CAMx domain on the west classified?

Fig 2: Is there a difference, conceptually, between blue vs orange vs black lines? If so, please clarify. If not, making them the same color may be an improvement (less distracting).

Fig 5: These types of scatter plots can be improved by adding color ranges to indicate the density of points.

---

## Author Comment (AC1) · 22 Aug 2017

Replies to Referees

Anonymous Referee #1 This manuscript applies a relatively new technique, the path-integral method, to characterize contributions to ozone over the United States. It applies two widely used models, CAMx nested within GEOS-Chem, to conduct its analysis. The main focus is on characterizing background and base ozone conditions, which have received heightened interest as the

U.S. has moved to tighter ozone standards.

The manuscript is clearly written and its methods are sound. Model performance and results are in line with previous studies.

What is relatively novel here is the use of the path-integral method. As such, additional explanation and illustration should be provided in the final paper. Specifically, in Figure 2, it would be helpful to illustrate what happens in the "PIM" box. Also, a bit more explanation would be helpful regarding Equation 1. In particular, it is unclear to me what is happening along the 4 steps – is each source being reduced 1/4 at a time?

Reply —- We have revised Figure 2 and its caption to clarify the source apportionment process. We also made some changes to Section 2.3 to improve the description of the method there. In Eq. (1), the sum is over the number of sources, now indicated as M, not the number of points at which the sensitivities are calculated. In our application, M = 4 and the 4 integrals are each estimated by a Gauss-Legendre integration formula using sensitivities calculated at three levels of emissions and BCs. This is described in the last paragraph of Section 2.3. The levels of emissions and BCs are not equally spaced but are determined by the zeroes of the 3rd order Legendre polynomial. The points and weights needed for Gauss-Legendre integration have been tabulated and are readily available. We have added a reference to the Gauss-Legendre integration formula that contains the points and weights. Further details of the PIM, including the transformation of the integration variable, are available in the earlier reference Dunker (2015).

Moreover, it is unclear to me that PIM has a "unique capability to allocate the difference in O3 between two simulations..." (p. 2, lines 27-28). Don't zero-out and tracer methods also do so? Given the nonlinearities of ozone response to emissions, there is no uniquely correct answer to apportionment. Better clarity on how to interpret PIM relative to other methods would be useful.

Reply — We clarified the statement on p.2, lines 27-28 to indicate that the unique capability of the PIM is that the sum of the source contributions is equal to the difference in O3 between the base and background simulations (within numerical error). As mentioned in the preceding paragraph of the manuscript, the sum of the source contributions from zero-out or tracer methods is not required to and usually does not equal the difference in O3 between the base and background simulations. I.e., the total anthropogenic contribution from the zero-out or tracer methods may over- or under-represent the actual contribution as determined by difference between simulations with and without the anthropogenic emissions. We also added a sentence at the end of the first paragraph of Section 2.3 stating that the sum of the PIM source contributions is mathematically required to equal the concentration difference between the two simulations. We agree that there is no unique source apportionment. However, because the focus of regulations is on the anthropogenic emissions, we feel that the PIM has an important advantage over other methods in requiring that the sum of the source contributions equals the anthropogenic O3 increment.

Soil NOx, lightning NOx, biogenic VOC, and wildfire emissions are all very important to background O3. It should be clarified how soil NOx was estimated, and whether each of these sources was modeled consistently between GEOS-Chem and CAMx.

Reply — We have added detail on sources of these emissions for the GEOS-Chem and CAMx simulations in sections 2.1 and 2.2, respectively. The data sources are different, in part due to differing requirements of the global and regional models, and we do not consider that consistency is required because CAMx is one-way nested within GEOS-Chem.

Finally, the authors could consider discussing the implications of their findings for the setting of ambient standards for O3, which is an urgent topic in the US now. That so many sites have base T10 concentrations above 60 ppb suggests certain limits under consideration may be unattainable in some regions. US NAAQS are formulated based on 4th highest O3 at the most polluted monitor in a region. Given that this study looks at T10 rather than 4th highest, at a single monitor rather than worst, with models that

tend to underpredict peak O3, and with the impossibility of removing all anthropogenic emissions globally all suggests attainment of NAAQS below 70 ppb may be unrealistic.

Reply — We agree that there are implications of our findings for setting an O3 standard below 70 ppb. However, given the focus of the journal, we do not think that it is appropriate to comment specifically on regulatory issues. We hope that US regulators or other interested parties will review our results and consider them alongside other scientific information in analyzing whether the current or lower standards are achievable.

Anonymous Referee #2 The manuscript by Dunker et al. estimates source contributions to the anthropogenic increment of O3 in the US. A one-way nested modeling framework combining GEOS-Chem and CAMx is used, allowing the authors to address global anthropogenic vs. background contributions. A relatively novel source attribution method, the path-integral approach, is used to alleviate some drawbacks of more traditional (brute-force, tracer) methods. The authors evaluate model performance as well as source attributions. The findings are useful, timely, and generally well explained and examined. Model performance is typical for these types of tools, and the authors explore reasons for discrepancies as well as alternative global modeling values. While some more work could be done to propagate this level of uncertainty into the final source attribution estimates, the overall technical approach is adequate. The introduction is a bit rough and needs some more work, the abstract is too vague, more rigorous and careful discussion and analysis is required to highlight the benefits of PIM, and there are a few other clarifying questions throughout on minor details. I believe in total these amount to minor revisions, and expect this article will be ready for publication in ACP without need for further review.

Main comments: 1.18-26: This is a good qualitative summary, but I would appreciate a more quantitative abstract. Meanwhile, with these very general descriptions, many of the statements are very obvious to readers familiar with the issue of background O3. For example, stating that contributions to background O3 from lateral BC's is largest for sites located near the boundaries seems quite obvious, although here one might

wonder if this statement truly applies to all sites or just those near boundaries with mostly inflow conditions. Other qualitative descriptions (largest, closest, larger, reduced, increases, increased,. . .) would significantly benefit from quantitative support. The abstract is not presently overly long and could easily be revised to contain such information.

Reply —- We lengthened the abstract and added quantitative details. We also revised the statement on the contribution of lateral BCs to the T10Base ozone at sites on the boundaries.

general: The introduction is rather limited. The writing style is curt and almost shorthand at times (see a few specific comments below); references and explanations are used somewhat casually. Overall, it reads like a first draft and would benefit from a much more polished presentation befitting the extensive experience of the authors.

Reply —- We have revised and somewhat expanded the Introduction and made changes in response to the specific comments below.

general: The authors perform some adequate model evaluation, and discuss reasons for different model biases. However, this sense of the magnitude of the model performance does not make its way, quantitatively, into presentation of the source attribution results. Would the authors be able to e.g. include some estimates of ranges in Table 3, or elsewhere?

Reply —- We do not know how to estimate the impact of errors or bias in model performance on the source contributions in Table 3 without greatly expanding the scope of the work. Probably the best approach would be to re-run the calculations with a different global and/or regional model, but this would require as much or more work than the results reported in the manuscript, depending on the alternative model(s) employed.

6.2. This is an interesting and useful finding. GEOS-Chem is a widely used model. Have previous studies made similar characterizations?

[Figure]

Reply —- We are not aware of other studies reporting a similar characterization. We obtained our finding by using two different thresholds in the analysis but other studies have apparently used just a single threshold.

6.30 - 7.4: I believe other studies have also indicated that lacking halogen chemistry (Br and Cl) can lead to high GEOS-Chem concentrations, as well as how isoprene nitrate species are recycled, or underestimation of O3 dry deposition. Later I see that halogen chemistry is mentioned in the conclusions.

Reply —- We added a sentence noting that lack of halogen chemistry may contribute to the O3 overpredictions. p. 7, line 3

Fig 7 and associated text: Indeed, background concentrations are higher than normal in an absolute sense when focusing just on days with the base case concentrations are high. But are they also higher in a relative sense? I would like to see another column to Fig 7 that shows the background values divided by the base for each row. Relative concentrations are currently only mentioned in this section for Denver, on line 10 of page 10. I can evaluate them myself for select sites using Table 3, but would appreciate more discussion be added.

Reply —- We added the requested plots as a new Figure 8 in the revised manuscript. We also added Table S8 to the Supplement, which gives the ratios of background to base concentrations for the 12 sites in Table 3, calculated using the spring, summer, T10Base and T10 Bkgd averages. We added a paragraph at the end of Section 3.4 discussing the new Figure 8 and Table S8 and also summarized the results in the abstract.

general: The title and framing makes me anticipate a bit more rigorous discussion of source attribution methods than what was included in this paper. The 6 specific points below address specific questions about methodology; in general though I wonder if the authors would really like to demonstrate the advantage of PIM if they would present their findings side-by-side with those from brute-force or tracer methods, both in terms

of the estimates of background concentrations and attributions and also the computational intensity (CPU and memory) of obtaining these estimates.

Reply —- We have expanded the Introduction to include more discussion of the PIM vs. the brute-force and tracer methods and responded to the specific points below.

2.23-26: While the description of the downsides of tracer methods is technically correct, I'm not sure the extent to which these are real limitations. For example, by how much do modern tracer methods not sum to the total increment? And while all nonlinear indirect reactions may not be included, to what extent are the most important ones included, which capture e.g. >90% of the O3 formation mechanism? A more practical downside, not mentioned here but which should be, is the computational burden of these tracer methods does not scale well when many sources (regions, times) are required, as the approach becomes memory intensive.

Reply —- Our view is that the PIM and tracer methods have different strengths and limitations, and that both are useful improvements upon the brute force method. We have expanded the discussion here, and in response to the next review comment, accordingly. Here, we added the following statement to illustrate how nonlinear chemistry may complicate interpretation of tracer method results: "If the chemistry changes significantly from the base to background cases, e.g., O3 production per nitrogen oxides (NOx) molecule becomes more efficient as NOx emissions are reduced, then an estimate of the anthropogenic increment using just the base-case chemistry can have important errors." The degree of this non-linearity varies, e.g., with emissions intensity and grid-resolution, which we believe precludes general statements about the accuracy for tracer methods. Our experience with memory requirements for tracer methods has been that CAMx can compute hundreds of source contributions in a single simulation and so we see memory footprint as an advantage of tracer methods rather than a limitation. We added the statement: "Tracer methods can be more efficient than both brute-force and the PIM as long as relatively few tracers (i.e., fewer tracers than chemical mechanisms have species) are used to perform source apportionment."

3.1: I feel like this statement is a bit unfair to PIM. The approach is more computationally intensive than a single brute-force sensitivity calculation. But PIM is efficient for obtaining the type of results that it is designed to calculate, compared to estimating the same contributions using other approaches. It also may be more CPU intensive whereas tracer methods would be more memory intensive. So, some more nuanced text here would be appreciated.

Reply —- Our view is that the PIM and tracer methods have different strengths and limitations, and that both are useful improvements upon the brute force method. We have added several sentences after p. 3, line 1 to expand the comparison between the PIM and the brute-force and tracer methods. In our previous study, Dunker et al. (2015), we found that the PIM required 2.7 times the computational effort of the brute-force method to apportion O3 and other species to five major source categories. Thus, the increased runtime for the PIM is a disadvantage. The PIM does produce concentrations (and sensitivities) at intermediate emission levels between the base and background cases, which is useful information on how pollutants change as emissions are reduced. This is now noted in the paragraph.

4.7: Particulates impact O3 via heterogeneous chemistry and photolysis. Many studies in the literature report these influences, which may be several ppb under particular conditions. Please explain further, quantitatively, and with references, why such effects are negligible in this case. My hunch is the authors will be forced to admit this decision was based, at least in part, on computational convenience, although I'm not explicitly sure why (does CAMx 6.3 not support DDM calculations for particulates? etc.).

Reply —- CAMx 6.3 does include DDM for PM. However, we agree with the referee that the influence of secondary PM formation on O3 will be only a few ppb or less, and that including PM in the simulations would significantly increase the computational effort. We confirm the reviewer's hunch by making this statement: "Formation of particulates was not included to reduce computational burden and because the focus was on O3"

5.4-6: There is no evidence provided to support the claim that this approach is "unbiased". It seems that in fact this assumption introduces a subjective bias in the analysis, which is the restriction that all precursor emissions change uniformly, which is clearly not representative of real-life conditions wherein emissions control measures target individual species (e.g., diesel NOx regulations). I think the rational here is one of simplicity and generality; if the study were more directed at the effects of particular control measures, different paths could be selected. One could also consider how emissions have changed, historically, and use those to define the path. But it doesn't appear that an analysis of emissions trends formed the basis of this statement, at least none is presented or cited.

Reply —- We revised this sentence to remove "unbiased." We also revised p. 5, lines 5-6 and added additional sentences about historical, long-term, national reductions of VOCs and NOx in the US. Although emission controls target specific sources and species (e.g., diesel NOx), we are focusing on large geographic regions (US, Canada and Mexico, and the remainder of the world) as sources. What is most important in this case is how the total VOC and NOx emissions in each of these large regions will change in the future. Our assumption seems reasonable based on long-term US history. An assumption based on detailed projections of worldwide emissions was beyond the scope of this work, and may not be possible for many countries outside the US.

5.11: Does this really account for the impact of all anthropogenic emissions on US O3? For example, if US anthropogenic NOx depleted some biogenic VOC concentrations, which then transported out of the US domain temporarily and then recirculated back into the domain at lower concentrations than would have occurred w/o these anthropogenic NOx emissions, would this be captured? I don't think so. At least this isn't mentioned on lines 5.14-15. I'm not arguing this is a substantial impact, quantitatively, just that the description of the method is neglecting some assumptions.

Reply —- If US emitted species and the secondary pollutants generated from them

move into Canada or Mexico, stay within the CAMx domain, and then return to the US, their impact is captured in our source contributions and assigned to the US. If the emitted species and secondary pollutants move outside the CAMx domain but then return to the CAMx domain via the BCs, the impact will also be captured, to the extent that GEOS-Chem has similar emissions, circulation patterns and chemistry as CAMx within the CAMx domain. What would happen in our calculations is that the effect of the recirculation would be ascribed to the lateral boundaries, not the US emissions. If the PIM were used with a single global model, the impact of the US anthropogenic emissions would be ascribed to the US, regardless of where the US emitted species and secondary pollutants are transported. Current tracer methods used in regional models estimate the contribution of the total BCs but do not separate out the contribution of anthropogenic pollutants arriving via the boundaries. The brute-force method could estimate the contribution of anthropogenic pollutants entering from the boundaries by difference between two simulations with different BCs. Again, the BCs would be derived from global model simulations with and without anthropogenic emissions. We are not aware, however, that the brute-force method has been applied to obtain such estimates of the contribution of anthropogenic pollutants arriving via the BCs. We have added a paragraph in Section 2.3 discussing recirculation of pollutants.

Section 3.4: The T10 metric is interesting, but it isn't clear to my why it was chosen. Are their findings sensitive to the use of 10 days rather than 3 or 30? Is there a policy relevance, like the 4th highest MDA8?

Reply —- The H4MDA8 has the limitation of being a single day that may not represent high ozone days in general. In averaging over 10 days, our goal was to include a reasonable variety of days with high total (T10Base) or high background (T10Bkgd) ozone over the 7 months of the simulation. Neither metric is directly related to the 4th highest MDA8 ozone, and we have not explored whether the results differ depending on the number of days that are included in the averages. There is some relevance of the T10Base average to policy in that calculation of relative reduction factors for

a nonattainment region by the U.S. EPA's procedure typically uses the top 10 days of MDA8 ozone. We feel that it is important to compare the T10Base and T10Bkgd results because they correspond to quite different transport, chemistry, etc. conditions. However, we also show source apportionments averaged over the spring and summer seasons (Figs. S10 – S13) and source apportionments for 23 sites over the full 7 months (Figs. 10, S13, S14). Although more analysis of results is always possible, the T10Base and T10Bkgd results along with the spring and summer averages and the time series at sites should give the reader a reasonably complete picture of the results. We have added a sentence in Section 3.4 explaining that T10Base is not directly related to a regulatory standard.

Minor comments / corrections: Title: Is CAMx really "nested" within GEOS-Chem in the usual sense of a nested grid model, or is it just using boundary conditions from GEOS-Chem (i.e., a one-way nesting)?

Reply —- The term "nesting" includes both one-way and two-way nesting. To make clear that we are using one-way nesting, we added "one-way nested" to the second sentence of the abstract, p. 1, line 11.

1.24: Not sure that a verb "increased" is appropriate here âËŸA ËĞT consider "higher".

Reply —- Changed "increased" to "greater." p. 1, line 24

2.1: "O3 background in the absence of anthropogenic emissions" is a vague phrase that needs to be more carefully written. A formal definition of what is meant by "background O3" in this particular manuscript should be clearly defined before this term is used. Further, assuming this has been defined, the rest of this statement seems redundant, for what other O3 besides the background would be present in the absence of anthropogenic emissions?

Reply —- Revised to "An important consideration is how difficult it will be to meet this standard by reducing US emissions alone because anthropogenic emissions outside

the US can contribute to US O3. This can occur by the transport of foreign anthropogenic emitted species into the US but more importantly by the transport of O3 and other secondary pollutants formed outside the US from the foreign emissions." Also, at the end of the next paragraph, stated that the focus of this work is on natural background O3.

2.3: Emissions are not transported, they are emitted. Species that are emitted are transported. Please tighten of the language in this regard here, and throughout.

Reply —- Changed to "emitted species." p. 2, line 3; p. 5, line 16; p. 11, line 9

2.3-4: Please provide a reference or references.

Reply —- Added three references.

2.8: Unless the acronym "NAB" is introduced here and used later, it doesn't make sense to capitalize Background.

Reply —- Made "background" lower case. p. 2, line 8.

2.13: the phrase in parentheses is missing some words in order to be grammatically correct.

Reply —- Changed to "known as the brute-force or zero-out method." p. 2, line 13

2.18: The chemistry âËŸAËĞT> Chemistry

Reply —- Deleted "The." p.2, line 18.

2.19: parenthesis not needed.

Reply —- Deleted parentheses. p.2, line 19

3.30: Use "CMAx" or "The CAMx model". Remove extra comma.

Reply —- Deleted "The"; removed comma. p. 2, line 30

Eq 1: This would be a more general equation if the "4" were replaced by e.g. M, where

M is defined to be the number of sources. Also, it seems implicit that the integral bounds are lambda=0 to lambda=1; it's not clear why "P" is used instead.

Reply —- Replaced "4" by "M" and added a definition of M. p. 4, lines 26, 28. Keeping P is important to keep the equation general because the path is not necessarily a straight line in lambda space. Using integral bounds of 0 and 1 may imply a straight line. To make this clearer, changed the last sentence of the paragraph to "P is some path from ðÌJę = 0 to ðÌJę = 1, not necessarily a straight line." Also, the standard mathematical representation of path or line integrals uses a symbol for the path unless the integral has been reduced to a definite integral incorporating the functions defining the specific path of interest.

4.28: An "array" is a computational object, not a mathematical one. Perhaps the authors mean "vector"?

Reply —- Changed to "vector." p. 4, line 29

5.1: This first sentence is confusing. What is the "direction" being referred to here? Some model time integration? How are "emissions added"?

Reply —- Changed to "In the integration direction of Eq. (1), emissions increase along the path as the $\lambda$m increase, . . ." p. 5, line 1

Fig 1: A small point of clarification âËŸAËĞT how are shipping emissions within the US Exclusive Economic Zone but outside the CAMx domain on the west classified?

Reply —- These emissions are in the global anthropogenic emissions used by GEOS-Chem but are not included in the CAMx simulation. Thus, they will affect the CAMx simulation via the boundary concentrations.

Fig 2: Is there a difference, conceptually, between blue vs orange vs black lines? If so, please clarify. If not, making them the same color may be an improvement (less distracting).

Reply —- In response to this comment and a comment by Referee #1, we revised Figure 2 and its caption to make the procedure for source apportionment clearer. The blue boxes and arrows now represent the flow of information to and from the basic model, CAMx. The orange boxes and arrows represent the flow of information to and from the PIM. Black lines have been removed.

Fig 5: These types of scatter plots can be improved by adding color ranges to indicate the density of points.

Reply —- We thank the reviewer for this comment and will investigate this option in future work.

---

## Author Response (AR2)

**Replies to Referee**

*Replies are in italics*

**Anonymous Referee #2**
page.line numbers refer to the revised manuscript with track changes

3.5-7:  For this particular case, couldn't it be more specifically stated that the anthropogenic increment would underestimated if just using the base-case chemical state?

*We revised the sentence to the following:*

*"If the chemistry changes significantly from the base to background cases, then an estimate of the anthropogenic increment using just the base-case chemistry can have important errors.  E.g., $O_3$ production per nitrogen oxides ($NO_x$) molecule becomes more efficient as $NO_x$ emissions are reduced, and therefore using just the base-case chemistry will very likely underestimate the anthropogenic increment."*

3.23:  It would be more general to say "changed" here rather than "reduced"

*Replaced "are reduced" with "change."*

[revised manuscript text omitted]